# Hygroscopicity of Waterlogged Archaeological Wood from Xiaobaijiao No.1 Shipwreck Related to Its Deterioration State

**DOI:** 10.3390/polym12040834

**Published:** 2020-04-06

**Authors:** Liuyang Han, Juan Guo, Kun Wang, Philippe Grönquist, Ren Li, Xingling Tian, Yafang Yin

**Affiliations:** 1Department of Wood Anatomy and Utilization, Research Institute of Wood Industry, Chinese Academy of Forestry, Beijing 100091, China; 2Wood Collections (WOODPEDIA), Chinese Academy of Forestry, Beijing 100091, China; 3College of Material Science and Technology, Beijing Forestry University, Haidian District, Beijing 100083, China; 4Wood Materials Science, ETH Zürich, 8093 Zürich, Switzerland; 5Laboratory for Cellulose & Wood Materials, EMPA, 8600 Dübendorf, Switzerland; 6Heritage Conservation and Restoration Institute, Chinese Academy of Cultural Heritage, Beijing 100029, China

**Keywords:** morphological structure, sorption behavior, sorption fitting model, compositional analysis, hydroxyl accessibility

## Abstract

Waterlogged archaeological wood (WAW) artifacts, made of natural biodegradable polymers, are important parts of many precious cultural heritages. It is of great importance to understand the hygroscopic behavior of WAW in different deterioration states for the development of optimal drying processes and choices of safe storage in varying conditions. This was investigated in a case-study using two *Hopea* (Giam) and two *Tectona* (Teak) WAW samples collected from the Xiaobaijiao No.1 shipwreck. The deterioration state of WAW was evaluated by the maximum water content (MWC) method and by the cell morphological structure. Both *Hopea* and *Tectona* WAW could be classified into moderately and less decayed WAW. The hygroscopic behavior of moderately and less decayed WAW was then comparatively investigated using Dynamic Vapor Sorption (DVS) measurements alongside two sorption fitting models. Compositional analysis and hydroxyl accessibility measurements of WAW cell walls were shown to correlate with the hygroscopicity of WAW in different deterioration states. It was concluded that moderately decayed WAW possessed higher hygroscopicity and hysteresis than less decayed WAW because of the lower relative content of polysaccharides and the higher relative content of lignin, including the slow hydrolysis of O-acetyl groups of xylan and the partial breakage of β-O-4 interlinks, accompanied by an increased hydroxyl accessibility. This work helps in deciding on which consolidation measures are advised for shipwreck restauration, i.e., pretreatments with specific consolidates during wood drying, particularly for wooden artifacts displayed in museums.

## 1. Introduction

Waterlogged archaeological wooden artifacts counting as valuable cultural heritages are being excavated worldwide [1,2,3]. In dependence of environmental factors, wood species, period, and processed treatments, WAW excavated even from the same archaeological site or collected either from the surface or the inner part of the same wooden artifact would probably be found in different deterioration states [4,5,6,7,8]. In general, WAW can be divided into severely decayed wood, moderately decayed wood, and less decayed wood [9,10]. The subdivision can be done according to maximum water content, morphological observations, and chemical structure of WAW [11,12,13]. Even though the water environment remarkably slows down the wood deterioration caused by microbiota, waterlogged wooden artifacts still suffer from a high possibility of deterioration. Hence, the conservation of WAW artifacts generally necessitates a water removal treatment [2,14]. 

Given that the two main forms of water in wood are free water in cell lumens and cell wall mesopores as well as bound water adsorbed in cell walls [7,15,16], the water-removal treatment incorporates both removal of free water and desorption of bound water. The treatment may lead to cell morphology changes due to surface tension that affect the dimensional stability of archaeological wooden artifacts because of cell wall shrinkage and collapse [15,17,18,19,20]. The sorption of bound water relies on the hygroscopic behavior of WAW, which is related to the dimensional instability that causes cracks and distortions in WAW [21]. Previous works have found that buried archaeological wood possesses a higher equilibrium moisture content (EMC) and higher hysteresis coefficients than recent wood [22,23], which could be attributed to the deacetylation of hemicelluloses, the degradation of amorphous celluloses, and a decrease of crystallinity. Moreover, WAW possesses varying deterioration states, each of which features an inhomogeneous deterioration behavior [24] and may require specific water-removal treatment to avoid unnecessary damage caused by cell wall shrinkage. Thus, it is of great importance to understand the hygroscopic response of WAW to different deterioration states in order to develop optimal drying processes, suitable display conditions, and safe storage under varying climatic conditions [25,26]. However, no studies yet illustrate the influence of deterioration state of WAW on its hygroscopicity.

The aim of this work is to compare the hygroscopicity of WAW in different deterioration states and to provide basic knowledge for their preservation, particularly in terms of the selection of drying method and storage conditions. Herein, two hardwood species of WAW, *Hopea* (Giam) and *Tectona* (Teak), were collected from the marine Xiaobaijiao No.1 shipwreck dated as 1821–1850 [27,28]. MWC and observation of cell morphological structure by LM and SEM were adopted to classify the deterioration state of WAW. The hygroscopic behavior was then examined by DVS, including analysis of equilibrium moisture content (EMC) and sorption hysteresis, and by fitting sorption/desorption curves with two frequently used multilayer sorption models; the Guggenheim, Anderson, and De Boer (GAB) and the Generalized D’Arcy and Watt (GDW) models. Furthermore, deuterium exchange measurements were conducted in order to gain information regarding the hydroxyl group accessibility. And finally, the study was complemented by compositional analysis in order to better understand the effect of cell wall degradation on the hygroscopicity of WAW. 

## 2. Materials and Methods 

The Xiaobaijiao No.1 shipwreck, now preserved in a waterlogged environment at the Ningbo Base of the Chinese National Center of Underwater Cultural Heritage (Ningbo, China), was a commercial ship in the period from 1821 to 1850 AD. The wreck site (scheme as shown in Figure 1) is located on Yushan Island, China [27]. Considering that an ideal selection of the reference wood specimens is a challenge for the comparison with the broader archaeological wood research, two samples corresponding to each of the two species composing the shipwreck were carefully chosen for comparable results. The locations of four samples (denoted H1, H2, T1, and T2) collected for this study are marked in Figure 1 and the detailed position information is listed in Table 1. Samples H1 and H2 were identified as *Hopea* spp. (Giam), while T1 and T2 were identified as *Tectona* spp. (Teak). As reported by a previous and related publication [28], the maximum water content (MWC) of the samples H1, H2, T1 and T2 were 121.59% ± 16.7%, 264.43% ± 80.55%, 108.51% ± 4.5% and 189.59% ± 65.36%, respectively.

### 2.1. LM

Cross sections of *Hopea* and *Tectona* WAW were prepared by a rotary microtome (RM 2255, Leica, Wetzlar, Germany) with a thickness of 10 µm. A light microscope (BX51, Olympus, Tokyo, Japan) was used to examine microstructure of the specimens.

### 2.2. SEM

Prior to the SEM examination, all cross-section surfaces of WAW were prepared by a rotary microtome (RM M2255, Leica, Germany). To avoid creating artificial damage to the cell wall structure during cutting with the microtome, the WAW samples were embedded with polyethylene glycol (PEG) 2000 (average molecular mass: 1900–2200 g/mol), which was followed by a rinsing process under flowing water for 30 min to remove the PEG. After mounting the dry samples on aluminum stubs followed by a sputter-coating process with Platinum, the samples were observed using a field emission scanning electron microscope (Quanta 200F FEI, Thermo Fisher Scientific, Waltham, MA, USA) at a voltage of 10 kV.

### 2.3. DVS

EMC of moderately decayed and less decayed WAW in different relative humidity (RH) states were measured by an automated sorption balance device (DVS Advantage ET85, Surface Measurement Systems Ltd., Wembley, UK). Measurements were mainly conducted according to the protocol found in [29]. Samples were cut into millimeter thick stripes by a razor blade and 30 mg of the waterlogged stripes were initially dried at a partial water vapor pressure of *p*/*p*_0_ = 0 for 600 min. The samples were then exposed to ascending p/p_0_ steps ranging from 0 to 0.98 for adsorption and then descending in the same manner for desorption at 25 °C. Equilibrium in each step was defined to be reached at a mass change per time (dm/dt) of less than 0.0005%/min over a 10 min stability window or a maximal time of 1000 min per step. The sorption hysteresis parameters were calculated by the difference of EMC for desorption and adsorption in the same relative humidity.

### 2.4. Isotherm Models

The obtained isotherms by DVS were fitted with two common sorption models, whose parameters were obtained by least-square fits to the data for each sample. The first of the models, the GAB model, was mainly improved from the commonly used BET isotherm model by increasing the sorbate activity range [30], and was first recommended by the European Project Group COST 90 [31] as a fundamental equation to characterize water sorption in food. It was subsequently introduced for the analysis of wood [7,32,33]. The GAB equation reads as:(1)$$\mathrm{EMC}=M_{m}\frac{K_{\mathrm{GAB}}\cdot C_{\mathrm{GAB}}\cdot \mathrm{RH}}{\left(1-K_{\mathrm{GAB}}\cdot \mathrm{RH}\right)\cdot \left(1-K_{\mathrm{GAB}}\cdot \mathrm{RH}+C_{\mathrm{GAB}}\cdot K_{\mathrm{GAB}}\cdot \mathrm{RH}\right)}\times 100\%$$
where *EMC* (%) is the equilibrium moisture content; *RH* (%) is the air relative humidity; *M*_m_ is the monolayer capacity; *C*_GAB_ (%) is the equilibrium constant related to the monolayer sorption, and *K*_GAB_ (%) is the equilibrium constant related to the multilayer sorption.

In addition, the internal specific surface area (*S*_GAB_) of WAW can be obtained based on the values of Mm provided by the GAB model:(2)$$S_{\mathrm{GAB}}=\frac{M_{m}\cdot \rho \cdot L\cdot \sigma }{M}$$
where ρ is the density of water, L is the Avogadro number, σ is the average area where water occupies the complete monolayer (0.114 nm^2^ was used in this study for the surface area occupied by a single water molecule) and M is the molar mass of water [26,34].

The second isotherm model used, the GDW model, assumes that the Langmuir mechanism governs the monolayer sorption, i.e., only one water molecule can be directly bound to a primary sorption site and that there are three possible scenarios: (a) the number of the secondary sites is lower than the number of primary sites (i.e., primary bound water molecules are not completely converted into the secondary sorption sites, *w* < 1. *w* is a conversion ratio of primary bound water molecules into the secondary sites); (b) the number of the secondary sites is equal to the primary sites (i.e., each monolayer molecule is converted into the secondary site, *w* = 1); (c) the number of the secondary sites is higher than the primary sites (i.e., each primary bound molecules creates more than one secondary sorption site, *w* > 1) [32].The GDW model [32,35] equation reads as:(3)$$\mathrm{EMC}=\frac{m_{\mathrm{GDW}}\cdot K_{\mathrm{GDW}}\cdot \mathrm{RH}}{\left(1+K_{\mathrm{GDW}}\cdot \mathrm{RH}\right)}\cdot \frac{1-k_{\mathrm{GDW}}\left(1-w\right)\cdot \mathrm{RH}}{\left(1-k_{\mathrm{GDW}}\cdot \mathrm{RH}\right)}\times 100\%$$
where *m*_GDW_ (%) is the maximum amount of water bound to the primary sorption sites, i.e., the monolayer water content. *K*_GDW_ (%) is a constant of sorption kinetics on the primary sites, and *k*_GDW_ (%) is a constant of sorption kinetic on the secondary sites. 

### 2.5. Compositional Analysis

The carbohydrates and total lignin of WAW specimens were measured with 3 replicates under the standard procedure according to the National Renewable Energy Laboratory (NREL, Golden, CO, USA) protocol [36,37]. Briefly, the milled specimens were hydrolyzed in 72% H_2_SO_4_ for 1 h at 30 °C and were then completely hydrolyzed in an autoclave at 121 °C for 1 h. The acid insoluble lignin was determined by weighing the solid, and the monosaccharides in the liquid were detected by high-performance anion exchange chromatography (Dionex ISC 3000, Sunnyvale, CA, USA).

### 2.6. Hydroxyl Accessibility 

The WAW samples measured by DVS were further used to study the hydroxyl accessibility. The samples, initially dried at *p*/*p*_0_ = 0 and 40 °C for 6 h, were exposed to D_2_O vapor at *p*/*p*_0_ = 0.95 and 25 °C for 10 h to ensure that the material’s accessible hydrogen protium is completely replaced by deuterium [38]. Then, the drying procedure was applied again, and the deuterated dry mass m_D_ was obtained. The number of available water vapor accessible OH groups (sorption sites) was calculated by equation (3) [29,39]:(4)$$\mathrm{Number}=\frac{m_{D}-m_{\mathrm{dry}}}{m_{\mathrm{dry}}\cdot \left(M_{D}-M_{H}\right)}$$
where *m*_dry_ is the dry mass of archaeological wood; *M*_D_ is the molar mass of deuterium and *M*_H_ is the molar mass of protium. 

## 3. Results and Discussion

### 3.1. The Deterioration State of Waterlogged Archaeological Wood 

According to the MWC values, the most commonly used parameter to classify the deterioration state of waterlogged archaeological wood [9,10], samples H2 and T2 belong to class of moderately decayed wood (185% < MWC < 400%), while H1 and T1 can be regarded as class of less decayed wood (MWC<185%). LM and SEM revealed the morphological structures of the WAW specimens and confirmed these deterioration states [6,9]. Cells in H1 mainly remained intact (Figure 2A) without significant deterioration features visible in the SEM image (Figure 3A). In contrast, the morphological structure of cell walls in H2 displayed pronounced decay patterns (Figure 2C). Parts of the S_2_ layers and the S_3_ layers of the fiber cell walls were degraded (Figure 3C), which is a sign of erosion bacteria decay pattern [24]. Furthermore, some parts of the cell walls of sample H2 were depleted by microbiological degradation with significant cavities emerging. Similar micro-morphological structure differences were present in T1 and T2. As for less decayed waterlogged *Tectona* (T1), the morphology shown by LM (Figure 2B) and SEM (Figure 3B) also indicated no pronounced sign of degradation. However, obvious features of soft-rot decay [24] were found in T2 by both LM (Figure 2D) and SEM (Figure 3D). Specifically, the S_3_ layers of the cell walls of T2 were still intact, while cavities occurred in its S_2_ layers. To sum up, the two moderately decayed WAWs were deteriorated to some degree by microorganisms with pronounced alternations of their cell morphologies, while the two less decayed WAWs in this research didn’t show any obvious cell morphologies indicating decay. Thus, the deterioration states of *Hopea* and *Tectona* WAW revealed by the morphological method were well consistent with the MWC method in this study. 

### 3.2. Hygroscopicity of WAW in Different Deterioration States

The sorption isotherms of *Hopea* and *Tectona* WAW in different deterioration states are shown in Figure 4 for a single adsorption and desorption cycle per sample. All adsorption and desorption curves display S-shapes, which implies that the sorption isotherms of both moderately and less decayed WAW might be classified as type IV IUPAC isotherms [40,41,42,43,44] (Whether the measured isotherms can be classified as type II or type IV can be debated. Some studies [41,43,44] classified wood sorption isotherms as type II, but this depends on the chosen interpretation of the sorption mechanisms; either exclusively mono/multilayer adsorption, or a mix with capillary condensation in mesopores of the cell wall [40]. Here the authors assume type IV.). The sorption isotherms of all specimens exhibited an upward bend at around 60%–80% RH, which is commonly reported in lignocellulosic materials [26,32,43]. Furthermore, the EMCs of moderately decayed WAW at each RH were all higher than those of less decayed WAW. At the highest relative humidity (98% RH), the EMCs of H2 and T2 reached as high as 25.91% and 27.37%, whereas the EMC of H1 and T1 reached 21.55% and 22.15%, respectively. Furthermore, different relative changes in EMCs (*M*_m_/*M*_l_) in both adsorption and desorption branches were present for both *Hopea* and *Tectona* WAW. The EMCs in both adsorption and desorption of H2 were at least 15% higher than those of H1 (Figure 5A). As for *Tectona*, the EMCs in adsorption for T2 were 19% to 39% higher than those of T1, and the EMCs in desorption for the former were 20%–49% higher than the latter (Figure 5B).

In addition to the increase of EMCs, sorption hysteresis of moderately decayed WAW was also higher than that of less decayed WAW. The sorption hysteresis is commonly calculated as the difference between adsorption and desorption branches of an isotherm in the range between the highest relative humidity (98% RH) and 0% RH [45]. It is believed to originate from a potential rearrangement of structural components in cell walls [46]. As shown in Figure 6, sorption hysteresis indeed exists in the whole moisture extent from 0% RH to 98% RH. The measurable sorption hysteresis of moderately decayed WAW was found higher than that of less decayed WAW at every humidity condition and for both *Hopea* and *Tectona* WAW. The higher sorption hysteresis of moderately decayed WAW as compared to less decayed WAW might lead to the higher variation in surface moisture content of wood elements under standard changes of relative humidity [46]. Therefore, conservators of archaeological artifacts usually try to consolidate WAW, for example, with lactitol and trehalose, in order to lower the sorption hysteresis [32,47].

In order to analyze the sorption process in detail, GAB and GDW sorption models were applied to fit the adsorption and desorption isotherms of *Hopea* and *Tectona* WAW with two deterioration states. The fits were considered to be valid if all the coefficient of determination (*R*^2^) values were above 0.99 [26,48]. The parameters calculated by a least-square fitting were listed in Table 2. 

With the analysis by the GAB model, as listed in Table 2, it could be noticed that the C_GAB_ values were approximately an order of magnitude higher than the *K*_GAB_ values for all samples, indicating much higher heat of sorption of the monolayer as compared to the multilayer [49]. It could also be deduced from the GAB model that the necessary condition for classifying the isotherms as type II (in this context, the authors assume that the sorption isotherm is the result of unrestricted mono/multilayer adsorption up to high p/p_0_) was satisfied because the conjunction of the relations 5.57 ≤ C_GAB_ < ∞ and 0.24 < K_GAB_ ≤ 1 was satisfied for the analyzed isotherms [32,50]. The maximum monolayer water content reflected by the *M*_m_ coefficient was found 11.74% higher for H2 than the *M*_m_ for H1. For *Tectona* WAW it was found 16.74% higher during the adsorption processes. For the desorption processes, the value for H2 was 8.80% higher than that of H1 and in the case of *Tectona*, it was 13.29% higher. The increased maximum monolayer water contents for moderately decayed WAW imply that long-time deterioration increased the number of accessible primary sorption sites compared to less decayed WAW. Besides, the *M*_m_ coefficient is proportional to the internal specific surface area (*S*_GAB_) [26,34], from which can be deduced that the internal specific surface area of WAW increases with the deterioration level. It was found that the *C*_GAB_ coefficient, which represents the total heat of sorption of the monolayer water [26,51], was higher for moderately decayed WAW than that for less decayed WAW. The C_GAB_ coefficients for H2 were 16.48% and 17.91% higher than those of H1, and in the case of *Tectona* they were 40.88% and 49.31% higher for both the adsorption and desorption processes respectively. These results could lead to the interpretation that the monolayer water is bound more strongly to the primary sorption sites for moderately decayed WAW as compared to less decayed WAW.

Using the GDW model, the maximum content of water bound to primary sites (*m*_GDW_) of H2 was 11.95% higher than that of H1, and in the case of *Tectona* WAW 34.77%, during the adsorption processes. During desorption processes, the m_GDW_ of the H2 sample was 43.50% higher than for H1, and in the case of *Tectona*, it was 38.97% higher. This could indicate an increased number of primary sorption sites for water in moderately decayed WAW. The ratio of water molecules bound to primary sites and converted into secondary sites (*w*) of WAW generally decreased with deterioration state. The values of H1 and H2 were the same. During the desorption processes, for *Tectona*, the value of T2 was 31.11% less than that of T1 during the adsorption. The value of H2 was 47.21% less than that of H1, and the in the case of *Tectona*, it was 35.79% less. The results would imply that each primary bound molecule of moderately decayed WAW created less secondary sorption sites than in the case of less decayed WAW. However, the decreased *w* of WAW in this study and other related studies [32,46] did not obviously contribute to the increased hygroscopicity because of the significant contribution of the increased number of primary sorption sites. 

The GAB and the GDW sorption models both indicated that moderately decayed WAW may possess more sorption sites and display a stronger capability of adsorbing water vapor from the surrounding environment than less decayed WAW. 

### 3.3. The Chemical Deterioration and Increased Hydroxyl Accessibility of Waterlogged Archaeological Wood 

The hygroscopicity of wood highly depends on the relative contents of the main components in the cell wall, i.e., cellulose, hemicellulose, and lignin [52]. The results of the compositional analysis of WAW in different deterioration states are shown in Table 3. The amount of cellulose, the most significant component, is reflected by the relative content of glucose. The content of hemicellulose can be assigned to the relative content of xylose, because in hardwoods, xylan dominates the composition of hemicellulose [52]. Finally, the Klason lignin content was used to reflect the content of lignin [53]. 

For H2, the relative content of cellulose counted by the proportion of glucose is 39.2% (according to total dry weight), 14% lower than that of H1. Xylose accounted for 3.9%, which was 47.3% lower than in the case of H1. In contrast, the relative content of lignin was 56.6% (55.6% Klason lignin and 1% acid-soluble lignin) for H2, 21% higher than that of H1. Additionally, it was found that the relative content of Klason lignin decreased from 55.6% to 46% from H2 compared to H1. As stated, the increased content of lignin in moderately decayed WAW does not indicate an absolute increase of lignin during the long-term deterioration process, instead, this is mainly a result from the loss of polysaccharides including hemicelluloses and cellulose [54]. For *Tectona* WAW, the compositional analysis presented the same tendency with a 9.6% higher lignin content and a 9.8% lower glucose content as well as a 0.8% lower xylose content for moderately decayed compared to less decayed WAW. 

The increase of EMC for the WAW can be understood in terms of the increase of both the number of sorption sites and their accessibility, as reported by a previous and related publication, where it was shown that the changes of chemical and cellulose crystallite structure led to an increased bound water uptake [7]. The hydroxyl accessibility of WAW was obtained in this work to further explore the apparent difference in hygroscopicity between moderately decayed and less decayed WAW.

Figure 7 displays the number of accessible hydroxyl groups obtained by sample deuteration using heavy water. The amount of accessible hydroxyl groups of wood generally ranges from 6.8–10.3 mmol/g according to previous research [38,55]. In this study, the number of accessible hydroxyl groups of H2 was 7.5 ± 0.4 mmol/g, while that of H1 was lower, with a value of 6.8 ± 0.3 mmol/g. For *Tectona* WAW, T2 showed a higher mean value of 7.9 ± 0.5 mmol/g, while T1 showed a lower accessibility of 7.0 ± 0.5 mmol/g. Because the amount of available OH groups is generally believed to directly correlate with the amount of sorption sites, the 10% and 13% higher numbers of hydroxyl groups in moderately decayed WAW could strongly indicate why the EMCs and hysteresis were higher in moderately decayed WAW than for less decayed WAW. 

The observed changes in hydroxyl accessibility could be either attributed to alternation of structure or to changes of chemical composition [29,56] upon deterioration. In fact, lignin, hemicellulose, and cellulose possess different amounts of hydroxyl groups [57], and as was reported above, they were each found to degrade to some different extent in WAW. The compositional analysis of different substrates revealed that the deterioration of WAW was mainly related to the decomposition of polysaccharides in the cell wall. Cellulose and hemicelluloses are in general extensively affected by anaerobic microorganisms and by occurring acid or alkali environment conditions [5] during the 170-years deterioration in waterlogged conditions. It is known that slow hydrolysis of O-acetyl groups of xylan always occurs in decayed archaeological hardwood [22,58,59]. As the hemicellulose is the most hydrophilic polymer in wood, its degradation generally results in a reduced hygroscopicity [60]. However, it should be emphasized that the hygroscopicity was higher for WAW than recent wood although the hemicellulose of WAW was severely degraded compared to cellulose and lignin, as illustrated by previous studies [7,32,47]. As shown in Table 3, the relative content of xylose representing hemicellulose decreased a lot. That means that besides hemicellulose, variations in the component structures of cellulose and lignin should also contribute to the increased hygroscopicity of WAW. Besides hemicellulose, cellulose is also considered as a major substance contributing to the hydrophilicity of wood and of other natural biodegradable polymers [61,62,63]. Although the interior of the cellulose microfibrils is not accessible to water vapor [64], the hydroxyl groups on surface chains are accessible [65,66,67]. Furthermore, the extent of moisture sorption of cellulose was reported to increase with decreasing degree of crystallinity [61,68]. In WAW, the deterioration of cellulose includes both the degradation of amorphous cellulose and the decrease of relative crystallinity [23,69]. In this research, the observed decrease of relative content of cellulose could demonstrate its degradation, and in addition, changes in amorphous cellulose and crystallinity were reported in the related previous study [69]. Therefore, both aspects might explain the observed increase in hygroscopicity. However, next to the deterioration of cellulose and hemicellulose, the modification of lignin may also lead to an increase of adsorbent functional groups in WAW [7,54,69], responsible for an increased uptake of water vapor. In detail, lignin in archaeological wood generally undergoes modifications such as the partial breakage of β-O-4 interlinks, an alternation of lignin structure, and demethylation/demethoxylation [7,54,70,71]. Although there is few studies on the relationship between WAW and the number of available OH groups, it was reported that the amount of available OH groups of delignified wood increased with the degree of delignification [29]. Both WAW and delignified wood possesses more OH groups because their lignin appears to be less bound to the carbohydrate matrix, enabling a flexible conformation. Additionally, the structure of lignin was partly altered, similar to extracted lignin [29,69]. In the current study, the increase in relative content of lignin of moderately decayed WAW (shown in Table 3) was caused by the loss of cellulose and hemicellulose, which implied that the lignin must have been modified to some degree throughout the deterioration in order to explain the increased hygroscopicity. To sum up, the demonstrated increase of hygroscopicity in more decayed WAW can be attributed to changes of relative components of the cell wall. It could be demonstrated that interpretations regarding hygroscopicity can correlate well between compositional analysis and hydroxyl accessibility measurements. 

## 4. Conclusions

Two hardwood species of waterlogged archaeological wood, *Hopea* and *Tectona*, were collected from a 170-year-old shipwreck, the Xiaobaijiao No.1, in order to investigate the effect of the deterioration on the hygroscopicity. Two deterioration states i.e., moderately decayed and less decayed WAW were confirmed by LM and SEM. In terms of water vapor sorption, moderately decayed WAW showed the higher EMCs and hysteresis than less decayed WAW. The fitted sorption models revealed that, towards WAW, the monolayer water was bound more strongly to primary sorption sites and the number of primary sorption sites increased with deterioration. The observed increase in hygroscopicity was attributed to a more severe deterioration of the wood chemical components of moderately decayed WAW in contrast to the less decayed WAW. In addition to higher hygroscopicity, moderately decayed WAW showed higher hysteresis than less decayed WAW, particularly above 60% RH, which is a domain close to museum humidity conditions (50%–60% RH for wooden artifacts in China). Therefore, considering the irreversible alternations in chemistry of WAW, the active reduction of sorption sites of WAW could be recommended as a possible strategy for conservation. Differentiated and appropriate conservation treatments would be required for WAW in different deterioration states.

## Figures and Tables

**Figure 1 polymers-12-00834-f001:**
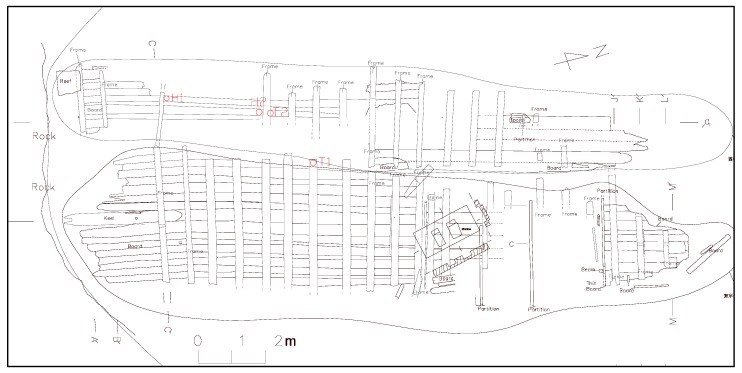
Scheme of the Xiaobaijiao No.1 shipwreck with the sample locations marked in red. *Hopea* WAW: H1, H2; *Tectona* WAW: T1, T2.

**Figure 2 polymers-12-00834-f002:**
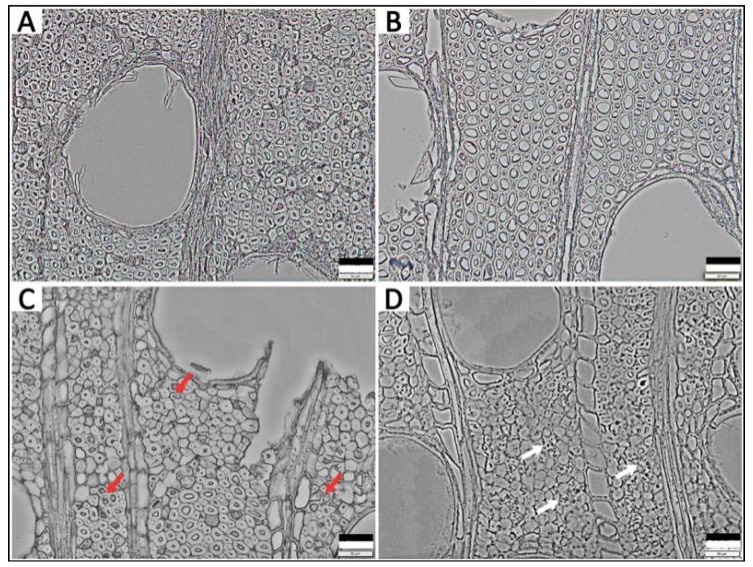
Light microscopy images of *Hopea* waterlogged archaeological wood (WAW): H1 (**A**): Less decayed sample, H2 (**C**): Moderately decayed sample (red arrows display pronounced decay patterns indicating degraded fiber cell walls); *Tectona* WAW: T1 (**B**), Less decayed sample, T2 (**D**): Moderately decayed sample (white arrows show that cavities occurred in its S_2_ layers indicate degraded fiber cell walls). Scale bar = 50 μm.

**Figure 3 polymers-12-00834-f003:**
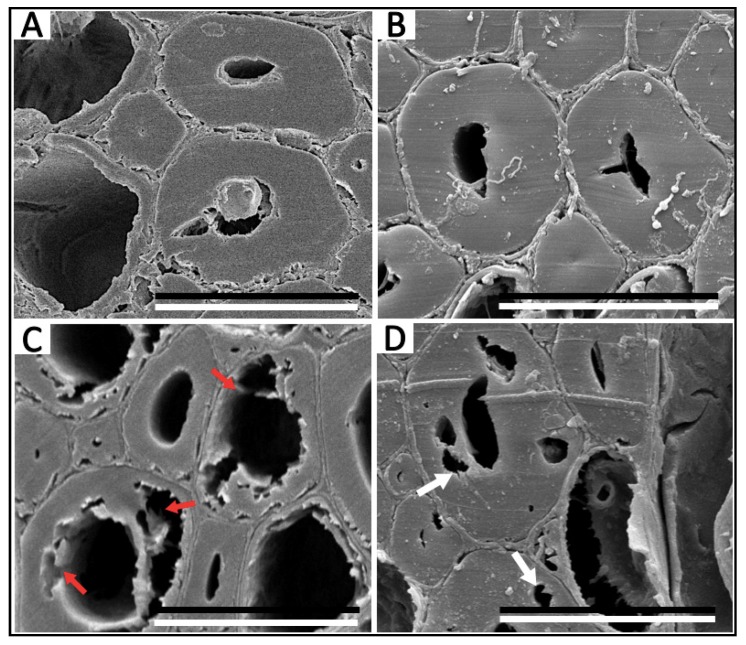
SEM images of *Hopea* WAW: H1 (**A**): Less decayed sample, H2 (**C**): Moderately decayed sample (red arrows indicate decay in the S_2_ and in many of the S_3_ layers within the fiber cell walls); *Tectona* WAW: T1 (**B**): Less decayed sample, T2 (**D**): Moderately decayed sample (white arrows indicate cavities visible in the S_2_ layers). Scale bar = 20 μm.

**Figure 4 polymers-12-00834-f004:**
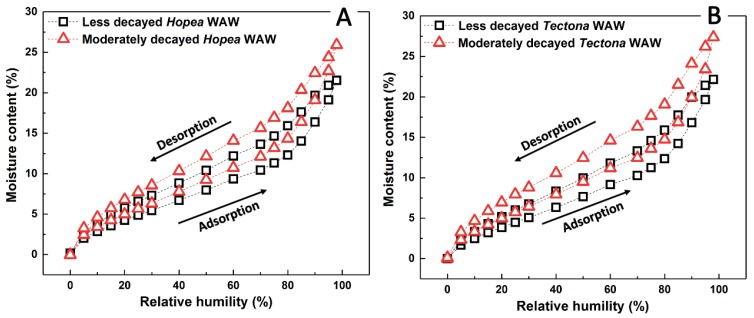
Equilibrium moisture content (EMC) of *Hopea* (**A**) and *Tectona* (**B**) WAW. Water vapor adsorption and desorption curves for the less decayed (H1 and T1, open black squares) and the moderately decayed (H2 and T2, open red triangles) samples.

**Figure 5 polymers-12-00834-f005:**
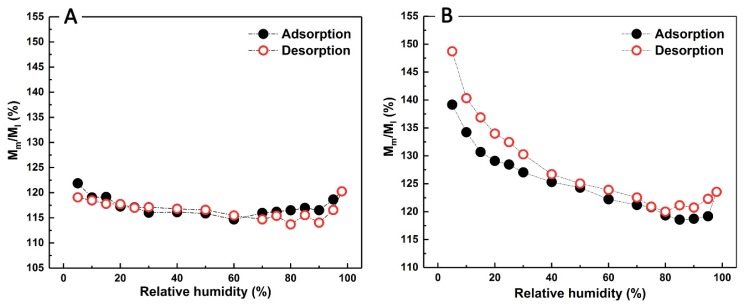
Relative changes in EMCs (*M*_m_/*M*_l_) in adsorption (black solid cycles) and desorption (red open cycles) of *Hopea* (**A**) and *Tectona* (**B**) WAW. *M*_m_ = moisture content (MC) of moderately decayed WAW (H2, T2); *M*_l_ = *MC* of less decayed WAW (H1, T1).

**Figure 6 polymers-12-00834-f006:**
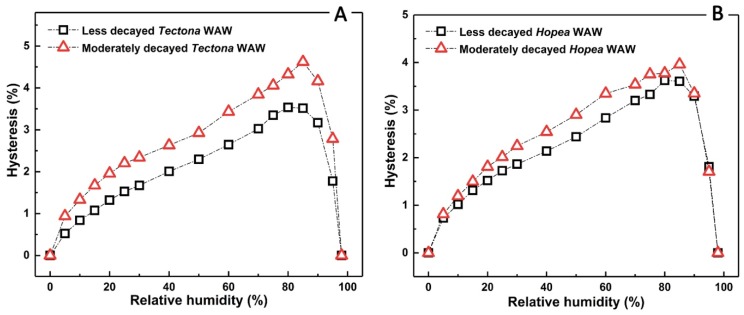
Sorption hysteresis of *Hopea* (**A**) and *Tectona* (**B**) WAW for the less decayed (H1, T1, open black squares) and the moderately decayed (H2, T2, open red triangles) samples.

**Figure 7 polymers-12-00834-f007:**
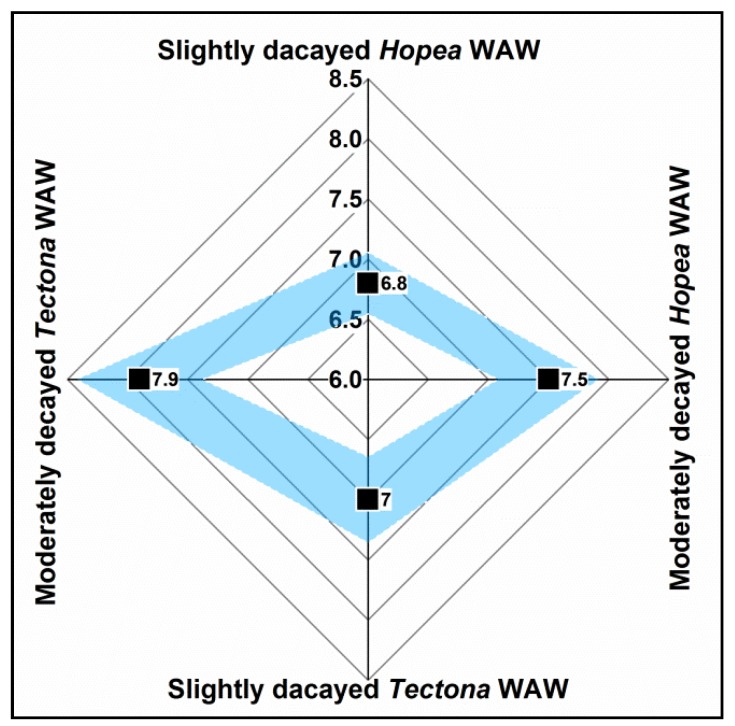
Radar image of the number (mmol/g) of accessible hydroxyl groups of the less decayed (H1, T1) and the moderately decayed (H2, T2) samples. The mean values (black squares) and the standard deviation (blue bands) were calculated from statistics for n = 3 measurements using representative samples.

**Table 1 polymers-12-00834-t001:** Detailed position of the samples collected from the Xiaobaijiao No.1 shipwreck.

Sample Name	Sampling Position
H1	The 5th inner layer board of the hull
H2	The 4th inner layer board of the hull
T1	The 6th inner layer board of the hull
T2	The 7th inner layer frame of the hull

**Table 2 polymers-12-00834-t002:** Coefficients of the GAB and GDW models for *Hopea* and *Tectona* WAW in different deterioration states.

Sample	Sorption Phase	GAB Model					GDW Model				
		*M* _m_	*K* _GAB_	*C* _GAB_	*R*2	*S* _GAB_	*m* _GDW_	*K* _GDW_	*k* _GDW_	*w*	*R*2
**H1**	Adsorption	4.94	0.79	15.59	0.999	187.68	8.70	4.41	0.86	0.39	1
	Desorption	8.30	0.66	10.72	1	315.34	6.69	11.32	0.58	1.97	1
H2	Adsorption	5.52	0.80	18.16	0.999	209.72	9.74	4.86	0.87	0.39	1
	Desorption	9.03	0.68	12.64	1	343.08	9.60	8.23	0.67	1.04	1
T1	Adsorption	4.84	0.80	11.79	1	183.88	8.57	3.71	0.85	0.45	1
	Desorption	8.20	0.67	8.01	1	311.54	6.62	8.38	0.61	1.90	1
T2	Adsorption	5.65	0.81	16.63	0.999	214.66	11.55	3.64	0.88	0.31	1
	Desorption	9.29	0.69	11.96	0.999	352.96	9.20	9.12	0.67	1.22	0.999

Note: *M*_m_ (%) is the monolayer capacity, *C*_GAB_ (%) is the equilibrium constant related to the monolayer sorption, *K*_GAB_ (%) is the equilibrium constant related to the multilayer sorption, *S*_GAB_ (m^2^/g) is the internal specific surface area, *m*_GDW_ (%) is the maximum amount of water bound to the primary sorption sites, i.e., the monolayer water content, *K*_GDW_ (%) is a constant of sorption kinetic on the primary sites, *k*_GDW_ (%) is a constant of sorption kinetic on the secondary sites, *w* – conversion ratio of primary bound water molecules into the secondary sites.

**Table 3 polymers-12-00834-t003:** Chemical compositions of WAW collected from the Xiaobaijiao No.1 shipwreck. Chemical composition 100% means related to investigated components. Statistics for n = 3 measurements. The standard deviations were less than 2%.

Sample	Acid-Insoluble Lignin	Acid-Soluble Lignin	Glucose	Xylose
H1	46.0%	1.0%	45.6%	7.4%
H2	55.8%	1.0%	39.3%	3.9%
T1	45.0%	0.8%	47.6%	6.6%
T2	54.6%	0.8%	38.8%	5.8%

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
