# Peer review of "Hygroscopicity of Waterlogged Archaeological Wood from Xiaobaijiao No.1 Shipwreck Related to Its Deterioration State"

_polymers, 2020, doi:10.3390/polym12040834_

Round 1

Reviewer 1 Report

Manuscript: Hygroscopicity of Waterlogged Archaeological Wood Related to its Deterioration State

Polymers-765006

Manuscript Presents very good research work related to Waterlogged Archaeological Wood Related to its Deterioration State cellulose material and required Minor revision before consideration to publication. Some minor comments are as follows.

  • Author need to incorporate some interesting structural characterization data in the abstract part of the manuscript. .
  • Authors need to incorporate some crescent references related to wood based material s and their application some examples are as follows;
    (a) Angew. Chem. Int. Ed. 2005, 44, 3358 – 3393 (b) materialstoday Volume, 21, Issue 7, September 2018, Pages 720-748 (c) Biomacromolecules 18 (8), 2333-2342 (d) ACS Sustainable Chemistry & Engineering 6 (3), 3279-3290 (e) Industrial & Engineering Chemistry Research 56 (46), 13885-13893 (f) ACS Sustainable Chem. Eng., 2019, 7 (6), pp 6140–6151 (g) Chemical Communications 49 (78), 8818-8820 (h) Cellulose 24 (12), 5417-5429 (i) Nanoscale, 2014, 6, 7764–7779 (j) ACS Sustainable Chem. Eng. 2018, 6, 2807−2828 (k) Environ. Sci.: Nano, 2018, 5, 623–658 (l) ACS Sustainable Chem. Eng. 2016, 4, 2632−2643 (m) Chem. Rev. 2018, 118, 11575−11625
  • Authors need to compare their results with previously reported similar results.
  • Author need to improve quality of figure 5 and 6.
  • Why Tectona, the EMCs in adsorption for T2 were 19% to 39% higher 209 than those of T1?
  • What method has been used to determine Acid-insoluble Lignin?

Author Response

Point 1: Author need to incorporate some interesting structural characterization data in the abstract part of the manuscript.

Response 1: Thank you for your comment. Some structural descriptions have been added in the abstract (lines 31-33) according to your professional suggestion.

Point 2: Authors need to incorporate some crescent references related to wood based materials and their application some examples are as follows;

(a) Angew. Chem. Int. Ed. 2005, 44, 3358 – 3393 (b) materialstoday Volume, 21, Issue 7, September 2018, Pages 720-748 (c) Biomacromolecules 18 (8), 2333-2342 (d) ACS Sustainable Chemistry & Engineering 6 (3), 3279-3290 (e) Industrial & Engineering Chemistry Research 56 (46), 13885-13893 (f) ACS Sustainable Chem. Eng., 2019, 7 (6), pp 6140–6151 (g) Chemical Communications 49 (78), 8818-8820 (h) Cellulose 24 (12), 5417-5429 (i) Nanoscale, 2014, 6, 7764–7779 (j) ACS Sustainable Chem. Eng. 2018, 6, 2807−2828 (k) Environ. Sci.: Nano, 2018, 5, 623–658 (l) ACS Sustainable Chem. Eng. 2016, 4, 2632−2643 (m) Chem. Rev. 2018, 118, 11575−11625

Response 2: Thanks a lot for providing us these meaningful and related researches. They are of great importance to our current work and are instructive for our future research! We are happy to cite some of them (Sharma, 2017; Eyley, 2014; Klemm, 2005) in this manuscript.

Point 3: Authors need to compare their results with previously reported similar results.

Response 3: Thanks for your professional comment. Comparative discussions have been added in section 3.2 and 3.3 under your kind suggestion.

Point 4: Author need to improve quality of figure 5 and 6.

Response 4: Thanks very much for your suggestion. The quality of related figures has been improved. In detail, the DPI resolution of all images has been improved from 300 to 600, and the pentagon markers in figure 5 have been repaced by cycles to enhance the visual aesthetic. Besides, the vertical scale of each image has been adjusted to a more suitable range as shown in the revised figures.

Point 5: Why Tectona, the EMCs in adsorption for T2 were 19% to 39% higher 209 than those of T1?

Response 5: Thank you for your question. In this research, the moderately decayed Tectona (T2) was deteriorated to some degree by microorganisms with pronounced alternations of its cell morphologies, while the less decayed Tectona (T1) in this research didn’t show any obvious cell morphologies indicating decay. Besides the morphological structure, the chemical composition of T2 showed the higher relative content of lignin and the lower relative content of hemicellulose and cellulose than those of T1. It is known that the water sorption of biomass is influenced by the structure and chemical composition [26], so the EMCs in adsorption for T2 were 19% to 39% higher than those of T1.

Point 6: What method has been used to determine Acid-insoluble Lignin?

Response 6: Thank you for your question. The method was according to the NREL standard [36,37]. Briefly, the milled sample was hydrolyzed in 72 % H2SO4 for 1 h at 30 °C, followed by a complete hydrolysis in an autoclave at 121°C for 1 h. The acid insoluble lignin was determined by weighing the solid, and the monosaccharides in the liquid were detected with a high-performance anion exchange chromatography. The related description was also added to the Section 2.5 in our manuscript.

Reviewer 2 Report

This manuscript presented a study about the hygroscopicity of two species of waterlogged archeological wood. The work is interesting. However, some points listed below, need to be improved.

Section 2.5: how many replicates were done in compositional analysis?

Section 3.3: please better discuss the results in this section. I suggest correlate the results of this work with others from the literature.

Page 10 Lines 294-295: the authors attribute the increase in EMC to the crystallite structure of wood. However, the authors did not determine the crystallinity of cellulose in wood. I understand that a relationship between the crystallinity cellulose structure with the water uptake can be done if the authors use XRD to obtain the parameters related to the cellulose crystallinity.

Author Response

Point 1: Section 2.5: how many replicates were done in compositional analysis?

Response 1: Thank you for your comment. 3 replicates were used for the compositional analysis. In the revised manuscript, it has been added in the section 2.5.

Point 2:  Section 3.3: please better discuss the results in this section. I suggest correlate the results of this work with others from the literature.

Response 2: Thanks a lot for your professional suggestion. More discussions (lines 355-358, lines 373-374, lines 404-410, and lines 422-427) have been added in section 3.3 in the revised manuscript.

Point 3: Page 10 Lines 294-295: the authors attribute the increase in EMC to the crystallite structure of wood. However, the authors did not determine the crystallinity of cellulose in wood. I understand that a relationship between the crystallinity cellulose structure with the water uptake can be done if the authors use XRD to obtain the parameters related to the cellulose crystallinity.

Response 3: Thank you for your suggestion. As speculated in page 10 Lines 362-364 (the line numbers in revised manuscript), the increase of EMC for the WAW can be related to the changes of chemical and cellulose crystallite structure according to one of our previous researches [7, 69]. It is understandable that the changes of chemical structure of wood will influence its EMC. WAW is a characteristic of the decreased crystallinity due to the wood deterioration, as illustrated by previous studies [7, 69]. In our recent wood [7], WAW had both the higher EMC and the lower crystallinity than those of recent wood (RW). In detail, the degree of crystallinity for the waterlogged archeological Idesia wood was 53%, which was lower than that of RW (59% for recent Idesia wood). The maximum EMC of waterlogged archaeological Idesia wood was almost 30% RH, however, that of RW was 21% RH. Thanks very much for your suggestion. We will pay more attentions on the relationship between EMC and crystallinity in archaeological wood as well as on the influence of wood species on this relationship in our future study.

Reviewer 3 Report

The research is reasonably interesting, although it has to be included in the category of “case study”, more correctly.

Since it does not present any innovation in the research, no new ideas, but only an application of a well known methodology. The article deals with a very topical issue but I'm not sure whether the content of the article is sufficient for publishing in this journal with such a high impact factor. In my opinion, the experiment need to be improved about the discussion of the results and to sensitize the readers about this variability of results.

Abstract

The abstract has too technical parameters. Please, reduce data and explain better the study to improve the comprehension of the work. I'll suggest to add better the aim of the study.

Keywords

Words from the title should not be used as keywords.

Lines 52-55. I invite the authors to rephrase the sentence.

Line 67. In scientific papers, the third person is favored.

Author Response

Point 1: The research is reasonably interesting, although it has to be included in the category of “case study”, more correctly.

Response 1: Thank you for your professional comment. We revised the title as “Hygroscopicity of Waterlogged Archaeological Wood from Xiaobaijiao No.1 Shipwreck Related to its Deterioration State” according to your kind suggestion.

Point 2: Since it does not present any innovation in the research, no new ideas, but only an application of a well-known methodology. The article deals with a very topical issue but I'm not sure whether the content of the article is sufficient for publishing in this journal with such a high impact factor. In my opinion, the experiment needs to be improved about the discussion of the results and to sensitize the readers about this variability of results.

Response 2: Thanks a lot for your professional comment. We revised this work according to your nice suggestions.

Waterlogged archaeological wooden artifacts with different deterioration states have been excavated all over the world. However, the characterization and information of WAW is still very lack to satisfy demand for its scientific preservation and protection. Understanding the hygroscopic response of WAW to their deterioration state is therefore important to develop optimal drying processes, suitable display conditions, and safe storage under varying climatic conditions [25, 26]. Herein, we selected Hopea (Giam) and Tectona (Teak) WAW samples with different deterioration states as a representative. The comparative analysis results could provide more information for the conservation of wooden artifacts.

Point 3: Abstract

The abstract has too technical parameters. Please, reduce data and explain better the study to improve the comprehension of the work. I'll suggest to add better the aim of the study.

Response 3: Thanks a lot for your good suggestion.  Sentences (lines 18-19 and lines 33-35) have been added to improve the comprehension of this work.

Point 4: Keywords

Words from the title should not be used as keywords.

Response 4: Thanks very much for your suggestion. Keywords have been revised into “morphological structure; sorption behavior; sorption fitting model; compositional analysis; hydroxyl accessibility”.

Point 5: Lines 52-55. I invite the authors to rephrase the sentence.

Response 5: Thank you for your professional suggestion. The sentence has been rephrased according to your comment (latest line numbers: lines 96-99 ).

Point 6: Line 67. In scientific papers, the third person is favored.

Response 6: Thanks a lot for your professional suggestion. The two related descriptions (including line 67) in section 1 and section 2 have been changed to the third person descriptions according to your kind reminder.
